# Impact of hospital accreditation on quality improvement in healthcare: A systematic review

**Mohammad J. Alhawajreh**[1]*, **Audrey S. Paterson**[2], **William J. Jackson**[2]

**1** Health Services Management, Zarqa University, Zarqa, Jordan, **2** Business School, University of Aberdeen, Aberdeen, Scotland, United Kingdom

* malhawajreh@zu.edu.jo, alhawajreh@yahoo.com

## Abstract

### Objective

This is the first systematic review aims to build the evidence for the impact of accreditation on quality improvement of healthcare services, as well as identify and develop an understanding of the contextual factors influencing accreditation implementation in the hospital setting through the lens of Normalisation Process Theory (NPT).

### Data sources

Data were gathered from five databases; MEDLINE, PUBMED, EMBASE, CINAHL, and the Cochrane Library. And supplemental sources.

### Study design

This systematic review is reported following PRISMA guidelines with a quality assessment. Data were analysed using a thematic analysis guided by the NPT theoretical framework.

### Data collection/extraction methods

Data were extracted and summarized using prespecified inclusion/exclusion criteria and a data extraction sheet encompassing all necessary information about the studies included in the review.

### Principal findings

There are inconsistent findings about the impact of accreditation on improving healthcare quality and outcomes, and there is scant evidence about its effectiveness. The findings also provide valuable insights into the key factors that may influence hospital accreditation implementation and develop a better understanding of their potential implications. Using the NPT shows a growing emphasis on the enactment work of the accreditation process and how this may drive improving the quality of healthcare services. However, little focus is given to accreditation's effects on health professionals' roles and responsibilities, strategies and ways for engaging health professionals for effective implementation, and ensuring that the

**Data Availability Statement:** All relevant data are within the paper and its Supporting Information files.

**Funding:** The author(s) received no specific funding for this work.

**Competing interests:** The authors have declared that no competing interests exist.

goals and potential benefits of accreditation are made clear and transparent through ongoing evaluation and feedback to all health professionals involved in the accreditation process.

## Conclusions

While there are contradictory findings about the impact of accreditation on improving the quality of healthcare services, accreditation continues to gain acceptance internationally as a quality assurance tool to support best practices in evaluating the quality outcomes of healthcare delivered. Policymakers, healthcare organisations, and researchers should proactively consider a set of key factors for the future implementation of accreditation programmes if they are to be effectively implemented and sustained within the hospital setting.

**Systematic review registration:** International Prospective Register of Systematic Reviews PROSPERO 2020 CRD42020172390 Available from: https://www.crd.york.ac.uk/PROSPERO/display_record.php?RecordID=172390.

## Introduction

Few, if any, countries around the world are satisfied with their healthcare systems. Over the past thirty years, improving the quality of patient care has become a top priority for healthcare organisations (HCOs) in both developed and developing countries [1, 2]. The Institute of Medicine (IOM) defines healthcare quality as *"the degree to which health services for individuals and populations increase the likelihood of desired health outcomes and are consistent with current professional knowledge"* [3].

The IOM (2000) revealed in its landmark report *"To Err Is Human"* that the majority of medical errors are caused by faulty systems and processes rather than by individuals [3]. As a result, the necessity for process improvement and safety initiatives in healthcare services has addressed the potential use of numerous quality improvement (QI) methodologies.

Improving healthcare outcomes through various strategies and quality improvement methods has become a global priority [4]. Researchers, health professionals, and policymakers are becoming more aware that healthcare accreditation programmes have the potential to improve the quality of healthcare services and support better performance for healthcare organisations. Accreditation is defined as "*a public recognition by a healthcare accreditation body of the achievement of accreditation standards by a healthcare organisation, demonstrated through an independent external peer assessment of that organisation's level of performance in relation to the standards*" [5].

While healthcare accreditation programmes have been used to improve the quality of healthcare delivered and patient safety in many healthcare organisations around the world, there are many concerns about their actual effectiveness [6, 7]. There is insufficient evidence to support the usefulness and efficacy of these programmes in improving the quality of healthcare services, and further research and exploration on its impact on quality improvement had been suggested [8, 9].

Hospital accreditation is not a simple solution. There are numerous difficulties in integrating it into routine work and maintaining its use [10, 11]. According to the accreditation literature, healthcare organisations face challenges and difficulties when attempting to use accreditation as a strategy to improve the quality of healthcare services [9, 12, 13]. As a result, developing a better understanding of accreditation implementation, including a clear clarification of the potential common drivers and barriers to a successful implementation process, could save significant amounts of time and resources.

This study provides a comprehensive systematic review to explore the impact of the hospital accreditation process on quality improvement of healthcare services; identify the contextual factors that might facilitate or hinder the effective implementation of hospital accreditation programmes; and provide a theoretical explanation of the impacts of hospital accreditation on quality improvement of healthcare services and the contextual factors using the theoretical framework of Normalization Process Theory (NPT). Two guiding questions direct the review: What is the impact of the hospital accreditation process on quality improvement of healthcare services? and what are the contextual factors that facilitate or hinder the effective implementation of hospital accreditation programmes?

In this systematic review, the accreditation literature is examined using the Normalization Process Theory (NPT) to underpin data collection and structure the synthesis. The NPT framework was chosen because it provides a structure to aid in the understanding, implementation, and evaluation of complex healthcare interventions, such as the accreditation process [14, 15]. NPT is a well-established middle-range theory concerned with explaining what people do to enable an intervention or innovation to become routine and normal components of day-to-day practise [16, 17]. The NPT has four core components or constructs for embedding of new complex intervention into routine practice. These four constructs are: Coherence, Cognitive Participation, Collective Action, and Reflexive Monitoring. Each construct represents different types of work that people perform during implementing a new complex intervention (see Table 1) [16, 18].

## Why it is important to do this review?

Previous reviews have attempted to assess the effect of accreditation on healthcare organizations performance, its relationship to specific quality measures and patient safety outcomes, common challenges to its implementation, and factors that might promote or inhibit its implementation in different contexts [7, 9, 19–22]. Since the last review was published almost a decade ago, the literature has grown rapidly and new important publications and developments need to be considered. Additionally, most previous reviews were methodologically flawed; they were non-systematic reviews and were not a rigorous review of evidence to determine whether hospital accreditation programs improve the quality of delivered healthcare services. Moreover, none of the previous reviews had used a theoretical framework to structure their synthesis. In response to these issues, we undertook an updated systematic review examining the implementation process of hospital accreditation and its impact of on the quality of healthcare services.

In contrast to previous reviews, this is the first comprehensive systematic review that conducted according to the current best practice guidelines and used the theoretical framework of NPT to review existing international literature on the accreditation implementation process

**Table 1. Normalisation Process Theory (NPT): Constructs and explanations.**

| Constructs | Explanation |
| --- | --- |
| Coherence | Sense-making: Do participants (individually and collectively) grasp the concept of a new intervention? |
| Cognitive Participation | Engagement: Do participants ªbuy into˚ a new intervention and seek to drive its implementation forward? |
| Collective Action | Enactment: Can participants enact the new intervention into practice in a real-world setting? |
| Reflexive Monitoring | Appraisal: Can participants evaluate the impact of a new intervention and generate ideas for reconfiguring practices to sustain its use over time? |

and the quality of healthcare services in hospital settings. This study therefore contributes to the NPT literature as well as hospital accreditation. This investigation will provide valuable insights into the complexities that healthcare organizations may face when implementing accreditation to improve healthcare quality.

## Methods

The systematic review followed the current best practise guidelines outlined in the Preferred Reporting Items for Systematic Reviews and Meta-Analyses (PRISMA) statement [23] (S1 File), and the Cochrane Handbook for Systematic Reviews of Interventions [24]. The study protocol was pre-specified and registered on PROSPERO 2020 (CRD42020172390). Data were analysed using a thematic analysis guided by the NPT framework. As this is a systematic review of published literature, no ethical approval was obtained.

### Eligibility criteria

All published articles in peer-reviewed journals were included if they met the study's eligibility criteria. The selection was limited to articles published in English to avoid any misinterpretation of the article's contents. The search was restricted to articles published between January 2009 and December 2022. The rationale for this time frame stems from our initial literature search into the impact of accreditation on the quality of healthcare services, which revealed either scant or inconsistent evidence prior to this point. Furthermore, prior to 2009, hospital accreditation was regarded as a relatively new and emerging field for health service research, particularly in developing countries [8, 25]. Finally, most previous studies have not addressed contextual issues that may have an impact on accreditation implementation [9].

### Information sources

Five databases were used as information sources (MEDLINE, PUBMED, EMBASE, CINAHL, and the Cochrane Library). A search of the websites of various national and international accreditation agencies, such as the Joint Commission International, as well as the included articles' reference lists were also searched to identify additional relevant studies (the snowballing technique).

### Search strategy

To avoid bias, the same search criteria were used across all databases whenever possible. Search results from databases and other sources were imported into RefWorks in order to remove duplicates from literature search results and to assess search results. The search strategy was revised and refined numerous times, with the complexity increasing in order to improve search results and obtain relevant search materials. The final search strategy was designed using the concepts in Table 2. An updated search was also conducted before the end of the study (on the 31st of December 2022) to identify any relevant newly published articles or any additional papers that were unavailable at the time of the initial search. Search strategies for two databases are available in (S2 File).

### Selection criteria and data collection process

Articles were assessed for eligibility using the PICOS criteria (Population, Intervention, Comparison, Outcomes, and Study Design) [26]; see Table 3 below. A complete description of inclusion and exclusion criteria is presented in Table 3.

**Table 2. Key concepts used as a framework for database searches on accreditation implementation.**

| Concepts | Alternative Key Words/Phrases | | | | | |
|---|---|---|---|---|---|---|
| **Concept 1** | hospital* | **or** | healthcare organization | **or** | patient care | |
| **and** | | | | | | |
| **Concept 2** | "accreditation*" | **or** | "certification*" | **or** | "licensure*" | |
| **and** | | | | | | |
| **Concept 3** | "Quality assurance, healthcare/" | **or** | "Quality improvement, healthcare/" | **or** | "Total quality, health care/" | |
| **and** | | | | | | |
| **Concept 4** | outcome | **or** | indicator | **or** | process assessment | |

The eligibility assessment for selecting articles was conducted independently by two reviewers using a standardised method. The study selection method was performed at three levels: the title was screened first, followed by the abstract and references, to identify articles for potential inclusion if they met the inclusion criteria. Where an abstract was not available from the database or it was difficult to make a selection decision on the basis of the title and abstract alone, the full-text article was accessed for screening. Each article found potentially eligible in line with the pre-specified inclusion criteria was read in full by the reviewers. At the full-text level, each article identified as eligible was assessed by two independent reviewers; any disagreements between reviewers about eligibility were resolved by discussion and consensus.

**Table 3. A complete descriptions of inclusion/ exclusion criteria for articles on accreditation implementation.**

**Inclusion Criteria**

1. Article published in the English language.
2. Article conducted in hospital settings (the **Population**).
3. Article discussed implementation of general accreditation and/or certification programmes of hospitals (the **Intervention**).
4. Article described non-accredited hospital, either by not participating in or not getting accreditation programs (the **Comparison**).
5. Article discussed the influence of hospital accreditation programmes on quality improvement, or identified the contextual factors influencing the effective implementation of hospital accreditation (the **Outcomes**).
6. Article evaluated the cost and financial impact of hospital accreditation.
7. Article comprised of any of the following study design (the **Study design**):
 7.1. Randomized controlled trials (RCTs) including both: (individual randomized controlled trials or group-randomized trial levels).
 7.2. Nonrandomized design, including:
 7.2.1. Controlled before-and-after study (CBAs) that included at least two locations in both control and intervention groups;
 7.2.2. Interrupted time-series study (ITSs) with at least three pre- and post-intervention measurements.

**Exclusion Criteria**

1. Article published in languages other than English.
2. Article was published in abstract format (complete study not available), editorial, letter, opinion, commentary, and conference reports.
3. Article where the population studied was non-hospital setting (e.g., primary healthcare organizations or other community-based healthcare organizations).
4. Article did not provide outcome data.
5. Article focused on single-specialty accreditation programmes (for example: oncology centre, radiology, surgical program, and medical education accreditation programmes).
6. Article comprised of any of the following study design:
 6.1. Randomised controlled trial (RCT) with only one intervention or control location;
 6.2. Non-randomised controlled trial (NRCT) including:
 6.2.1. Controlled before-and-after study (CBAs) that included with only one intervention or control location;
 6.2.2. Interrupted time-series study (ITSs) that do not have a clearly defined point in time when the intervention occurred and at least three pre- and post-intervention measurements.

A data extraction sheet based on the Cochrane Consumers and Communication Review Group Data Extraction Template (S3 File) [24] was used to extract and summarise raw data from included articles. Two reviewers extracted data separately. Any disagreements were resolved by discussion and, where necessary, consultation with a third reviewer.

In order to minimise bias and maximise generalisability, the study design was documented according to the Cochrane Handbook guidelines [24] as a (RCTs) design, including both: (individual randomized trials or group-randomized trial levels) or as (NRCTs) design, including: (ITSs), (CBAs) or uncontrolled before-and-after studies) or controlled observational studies (cohort studies, case-control studies, cross-sectional studies, and case series). The previous order represents the hierarchy of methodological rigour of studies.

## Synthesis of results

A narrative synthesis of the results using thematic analysis was undertaken with the consideration of assessing the methodological rigour and quality of the included articles. Due to the insufficient number of experimental studies and the substantial heterogeneity within the included articles in terms of the research methods used (a wide range of qualitative, quantitative, and mixed-method studies) and outcome measures, quantitative meta-analysis of the data was not possible. The NPT was employed as a theoretically informed coding framework to structure the data synthesis and guide the assessment of contextual factors influencing the implementation process of accreditation reported in the reviewed articles (S4 File).

## Coding of studies to NPT

For each of the 21 articles included in the qualitative analysis; review data on the facilitators and barriers to, and the impact of hospital accreditation on quality improvement of healthcare services was extracted, summarized, and coded to the related construct of the NPT framework using a Microsoft Excel Worksheet (S5 File). If review data about the accreditation implementation process could not be connected to the NPT, this was defined to make sure that information beyond the framework of the NPT would still be captured. The coding and interpretation process of review data focused mainly on the results and discussion parts of each included article. This work was undertaken independently by two reviewers. To ensure accuracy and uniformity of the analysis and coding process, the reviewers then conducted multiple meetings to review coding decisions. Any discrepancies were resolved through discussion and by consulting a third reviewer where needed to reach consensus.

## Assessment of included studies

Since the included articles did not encompass enough data from the RCT studies, the risk of bias tool outlined in the Cochrane Handbook was considered unsuitable for assessing the risk of bias. The quality of included articles was assessed using a quality assessment method, the Quality Assessment Tool developed by Hawker et al. [27], which is designed to critically appraise studies from a wide range of research designs to ensure consistency and strength in the quality evaluation process. The assessment tool comprises nine categories, which are based on a point system that ranges from one to four for each of its nine categories: one point (very poor), two points (poor), three points (fair), or four points (good). As a result, each article received a score of a minimum of 9 points and a maximum of 36 points based on the study appraisal criteria (S6 File). Quality scores were performed by two reviewers, with discrepancies resolved through consensus.

## Results

### Study selection

Searches of the databases identified 1345 potentially relevant citations, and 36 additional citations were retrieved through other search methods. Of these 1381 citations, 453 were duplicate hits, which were removed from further screening. After reviewing the titles and abstracts of the remaining 928 citations, a further 775 were excluded as they did not match the inclusion criteria. The full text of the remaining 152 articles was reviewed in detail, resulting in another 131 being excluded. The remaining 21 articles met all eligibility criteria and were included in the final review. The systematic review screening and reasons for exclusion process is illustrated in a PRISMA flow diagram [28], Fig 1. The included and excluded studies and reasons for exclusion are presented in (S7 File).

### Study characteristics

Of the remaining 21 articles, no articles were excluded after conducting the quality assessment since none were found to be at a low-quality level. In total, only one article was interventional-a randomized controlled trial (RCT) [29], and the remaining were observational, non-randomized designs. The majority of the observational studies were categorised as cross-sectional. Of the articles identified in the final review, nine used quantitative design, eight employed qualitative design, and four involved mixed methods design. The included articles covered 11 countries; the largest number of articles were conducted in Australia, Jordan, and Iran (n = 3, 4, and 3 respectively); two articles were carried out in each of the following countries: Canada, Denmark, and the USA; and the remaining five articles were conducted in Lebanon, Portugal, Saudi Arabia, Turkey, and the United Arab Emirates.

All studies were conducted in hospital settings, including public, private, teaching, accredited, and non-accredited hospitals. Most of the articles comprised diverse stakeholders such as patients, physicians, nurses, and healthcare managers, focusing on analyses of hospital documents and quality measures related to patients' outcomes and hospital performance. While each article had a specific aim and research question, all of them discussed the implementation of general accreditation and/or certification programmes and the impact of accreditation on various hospital performance measures. Across the articles, the focus was on the influence of hospital accreditation programmes on quality improvement or the contextual factors influencing the effective implementation of hospital accreditation. Of the 21 selected articles, only 2 articles reported on the unintended consequences of hospital accreditation programmes [29, 30]. The effect of hospital accreditation on cost was rarely discussed (unless the aim of the article was to assess the economic impact of accreditation).

### Results of quality assessment

In this review, each article was awarded a total quality score and classified based on the following quality grading system: high quality (30–36 points), medium quality (24–29 points), and low quality (9–23 points). Quality scores ranged from 25 to 36 across the 21 included articles. Overall, the selected articles were found to be either of high or medium quality standards, and therefore all 21 articles were included in the final review. A summary of the 21 articles included in the final review and the quality assessment results are presented in (S8 File).

### Main results

As per the research objectives, the focus was on identifying a priori themes using the NPT. However, the data analysis involved two stages. First, data were analysed following the

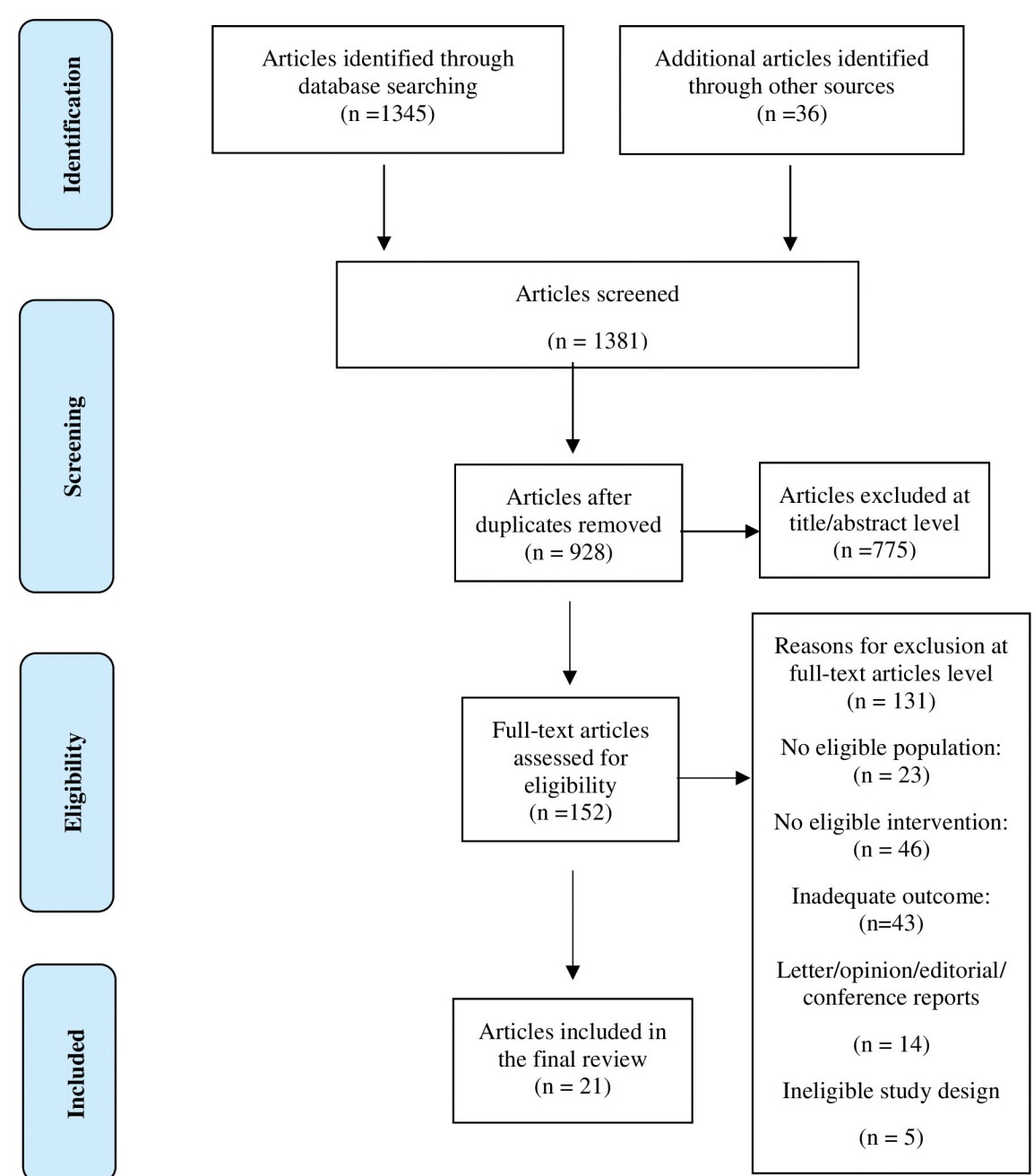

**Fig 1. This is the Fig 1 PRISMA flow diagram showing the number of articles identified in the literature search and the study selection process.**

principles of a framework thematic analysis approach using the framework approach provided by Ritchie and Spencer [31]. In stage two, in order to develop a better understanding of the implementation, embedding, and integration of the accreditation, the emerging themes were then guided by the NPT framework. The key contextual factors that were found to influence the implementation of accreditation were also identified within the reviewed articles and summarised in Table 4.

The reviewed articles yielded a wide range of topics and ideas. These topics were then grouped into themes and sub-themes and mapped against the four NPT constructs below,

**Table 4. A summary of the main contextual factors affecting accreditation implementation organised according to the four constructs of the NPT framework.**

| NPT construct | Factors affecting implementation |
|---|---|
| Coherence | • Accreditation program is fitting with healthcare organization vision, beliefs, and principles.<br>• A collective perception for the specific objectives, value, and potential benefits of accreditation across diverse healthcare professionals.<br>• Adequate knowledge and awareness regarding health professionals' specific roles and responsibilities in the implementation of accreditation process.<br>• Healthcare professional understand that accreditation is a new quality improvement technique, but complementary to current methods and practices.<br>• Training sessions and workshops to clarifying the standards of accreditation program. |
| Cognitive Participation | • Continuous education and training sessions about implementing the standards of accreditation.<br>• Technical assistance and support from accreditation bodies, specialists and key persons.<br>• Healthcare organization leadership commitment, support and engagement.<br>• Adequate financial support and proper equipment.<br>• All health professional groups are motivated to be involved in the accreditation program (particularly physicians and senior managers).<br>• Role identity, all health professional groups convinced that accreditation should be part of their role. |
| Collective Action | • Continuous education and training sessions about implementing the standards of accreditation.<br>• Skills, performance, and characteristics of health professionals involved in the implementation of accreditation programmes.<br>• Building confidence and trust among health professionals and in accreditation process.<br>• Facilitating communication and encouraging knowledge share and teamwork.<br>• The negative impact of increasing workloads and time pressures, which result in a degree of a role conflict for health professionals (particularly nurses and physicians).<br>• Sufficient budgets and adequate resources, including (financial, human, and social resources).<br>• Readiness of healthcare organizations for implementing accreditation program. |
| Reflexive Monitoring | • Health professional's participation in the adjustment and revision of the accreditation standards to fit the context (from feedback).<br>• Introducing financial incentives, rewards and appropriate recognition.<br>• Adapting the healthcare organizational culture.<br>• Clarifying the roles and responsibilities for each participant in the accreditation process.<br>• Including accreditation measures in health professional performance evaluation.<br>• Health professionals perceive the accreditation value as a useful tool to improve the quality of care (Positive feedback). |

which are supported with illustrative quotes from the included articles' review data. The final examined themes were as follows:

- The impact of accreditation on hospital performance and quality improvement measures, which include:

✓ The accreditation process is associated with positive or negative changes in hospital performance and quality outcomes.

- Key contextual factors affecting the implementation of hospital accreditation include:

✓ Factors identified as critical enablers or barriers to effective accreditation implementation.

## Coherence (sense-making)

In the context of this study, sensemaking involved how healthcare professionals view the hospital accreditation process as a new way of working, how it compares to existing practice, their shared understanding of its aims and objectives, their specific tasks and responsibilities in the

implementation of accreditation, their perceptions of how it affects them individually, and its likely costs and expected benefits.

The key findings of the coherence construct reveal that hospital accreditation differs from existing healthcare delivery models in that it is regarded as a strategy for positive change and a powerful tool for standardising healthcare services in hospital settings and improving healthcare quality. It also indicated that the implementation of accreditation standards increased recognition (communal specification) among healthcare professionals, who learned that they were responsible not only for providing routine care but also for ensuring that care was safe and aligned with the approved set of standards. individually, as well as its likely costs and anticipated benefits.

> "*Another aspect associated with the higher formalisation fostered by the accreditation process was the development of performance management systems. . . Besides contributing to a reduction of the silo mentality, the new performance management systems also improved staff's motivation to contribute to quality and patient safety improvements, including the notification of patient safety incidents.*" [32] (p.1249).

However, while healthcare professionals perceive accreditation as a technique for providing good healthcare practices, they do not collectively share and create a clear understanding of its specific objectives and potential benefits. For example, Hijazi et al. [2] (p.12) reported that:

> "*Administrator staff and health care professionals (e.g., physicians, pharmacists, and therapists) do not perceive that developing a [Quality Assurance] QA framework and supportive policies [e.g., Accreditation] is important for patient centeredness. . .[However], nurses reported that establishing a QA infrastructure that values patient goals and needs could be a method to provide care that is ultimately focused on the patient.*"

From an individual perspective, healthcare professionals demonstrated a good understanding of the main goals of accreditation. However, there were concerns regarding their limited knowledge and insufficient awareness of their specific tasks and responsibilities in the implementation of accreditation, which could trigger negative feelings and resistance. Training sessions and workshops to educate staff on their roles in the accreditation process were deemed necessary to ensure the program's smooth implementation.

> "*Most of the respondents agreed that their knowledge of the new standards and practices for quality was very limited when they started the process. Because of a lack of knowledge and skills, some staff were uninterested or uncertain about how to introduce these concepts into the hospital.*" [33] (p. e272).

> "*. . . It should be better at first to employ manager of hospital according to job qualification and criteria which is needed for management of hospitals. And also prepare training courses related to accreditation implementation for them to increase their knowledge and skills.*" [34] (p.52)**.**

While there is evidence that individuals understand the value, benefits, and importance of the accreditation, some health professionals were in doubt if it would lead to a significant benefit, which affected their acceptance (internalisation) of the new standards.

> "*Participants held that health professionals' comprehension of the utility and value of accreditation modifies their engagement in programs. Professionals were characterised as often*

*harbouring doubts about the ability of accreditation to promote organisational and health system improvements".* [35] (p.5).

## Cognitive participation (relationship work)

The importance of supporting health professionals to actively participate (enrol) in preparing the healthcare organization for the accreditation process and to establish organizational buy-in to the accreditation featured highly in the articles. Time, or lack of, heavy workloads, and staff shortages were noted as common factors in the slow implementation of the accreditation.

Furthermore, some articles revealed that health professionals in some organisations attempted to persuade their colleagues to participate in the accreditation standards but were unsuccessful. This was put down to a lack of staff awareness about accreditation objectives and its implementation requirements, or to the fact that not all health professionals received enough education and training prior to the accreditation process. In general, scepticism over the value of accreditation and its potential benefits made it challenging to get them actively engaged in the implementation process.

> "*The accreditation process took place just a few months after the merger and was conducted by nurse managers who were also in charge of quality improvement. Doctors' participation varied by self-assessment group, but overall, doctors did not much participate.*" [36] (p.8).

From an activation perspective, support from accreditation specialists and key persons was identified as an important factor in improving conditions to take forward the implementation process of accreditation and to become part of everyday work.

> "*. . . strong relationships with accreditation staff and surveyors played a critical role in providing the support required to create a consistent approach across the organization. When strong relationships were not present, organizations were less likely to achieve buy-in.*" [37] (p.944).

The importance of training was also noted, where designated individuals such as organisational leaders, accreditation specialists, and surveyors would communicate, encourage, and instruct their colleagues, as well as motivate other health professionals, such as physicians, to take part in the accreditation and become involved.

> "*The literature also indicates that for training to be effective, it needs to be sufficient, continuous, well designed and well delivered, demonstrably relevant to day-to-day activities and focused on equipping individuals with the understanding and tools to improve the quality of health care.*" [33] (p. e276).

Individuals' willingness and efforts to initiate the implementation of accreditation and their ability to engage others in the implementation process were noted as important.

> "*The operational champion is responsible for liaising with the accrediting body and overseeing the individuals and processes involved in preparation for the on-site survey. Individuals in this role were described as actively supporting the progress of the organization's quality agenda; understanding the meaning behind accreditation standards and applying it to the context of the organization; and communicating the meaning behind the standards to individuals and departments across the organization, ensuring a consistent approach.*" [37] (p.944).

However, it is also recognised that while individuals can be the drivers (initiators), they can also be barriers to the implementation process.

In general, health professionals were relatively convinced and comfortable that it was legitimate for them to engage in the hospital accreditation process. However, there was some variation in participation among health professionals from across disciplines; for example, mixed views were reflected in some included articles among physicians, nurses, and senior managers. Ehlers et al. [38] (p.697) commented that:

*"The attitudes towards accreditation and [the Danish Quality Model] DDKM were in fact supportive. Typically, hospital physicians are more sceptical than others, but nevertheless they largely affirm accreditation's positive effect on organizational quality. Nurses, managers, quality improvement staff and surveyors hold positive attitudes. Only a small group of physicians was extremely negative."*

On the other hand, in some healthcare organizations when physicians were engaged in the accreditation process, particularly when they played the role of local champion for the accreditation, this was considered a strong lever for promoting the implementation process forward.

*"It is often claimed that the participation of doctors and nurses and ensuring their motivation are critically important in order to put accreditation standards into practice."* [39] (p.78).

## Collective action (enacting work)

A large amount of data was discussed and described as to how health professionals worked to embed the new set of accreditation standards with already known and existing practises of routine healthcare to ensure effective delivery of daily work. Throughout the included articles, several key factors have been identified that influence the health professional's ability to implement the accreditation, including financial support, human resources, proper equipment and tools, time, staff skills and experience, and staff motivation.

From a skill-set workability perspective, the main concerns relate to allocating resources and assessing the necessary skills of the individuals involved in the accreditation implementation process. Many articles revealed that accreditation was associated with a positive change in hospitals in terms of increasing attention to infection control issues, enhancing health professionals' performance, encouraging teamwork and knowledge sharing, and developing individuals' skills in the application of continuous quality improvement activities. In this respect, Melo [32] (p.1250) reported that:

*". . . Although the accreditation process was managed by a team of staff of the quality management department, several thematic working groups were created in order to develop specific procedures. Each working group was formed by staff from several departments across the hospital who were actively involved with the topic in question in their day-to-day activity."*

Conversely, the extra time and effort needed to do the tasks and paperwork related to accreditation was a barrier to performing their essential work of providing routine care, consequently increasing demands on their time and reducing time spent directly on patient care.

*"Staff shortages remain one of the biggest challenges facing accreditation. Half of the respondents highlighted that while their departments already experienced staff shortages, the accreditation process required additional staff to deal with the amount of paperwork."* [33] (p. e270).

Organizational support in the form of resources to enable the contextual integration of the accreditation program to embed accreditation standards into practice featured highly in most of the studies. Likewise, senior manager support, commitment, and interaction with team members were considered essential in helping health professionals actively participate in accreditation implementation.

"*Two interrelated attributes of healthcare organisations were proposed to enable effective implementation of accreditation programs: credible leaders that champion continuous quality improvement (CQI) and the role of accreditation; and organisational cultures which promote collective staff ownership for CQI.*" [35] (p.5).

The findings revealed conflicting views on the actual effectiveness of accreditation in terms of the level of confidence that individuals have in the accreditation process and in those who implement it. Numerous health organisations around the world have implemented accreditation as a strategy for improving the quality of care and patient safety. While many health professional groups believed such a strategy was ineffective.

"... *To examine the impact of accreditation on nurses' perceptions of quality of care. Unlike studies in other countries... our study results ... showed that implementing accreditation has no positive impact on quality results.*" [40] (p. e236).

Others perceived that it helped build confidence and trust among health professionals by facilitating communication, encouraging teamwork, and strengthening relations between health organisations and other stakeholders.

"*Many respondents believed that the [accreditation program] had contributed significantly to the improvement of communication, through the availability of policies and procedures that facilitated greater unity among members of hospital staff.*" [33] (p. e272).

## Reflexive monitoring

Monitoring and evaluating accreditation implementation are critical for understanding how it affects health professionals and other stakeholders. Health professionals described making reconfigurations to the accreditation process to fit the context as a critical factor for its effective implementation. They also highlighted the role of key persons, such as surveyors and accreditation consultants, who were necessary to help facilitate adaptation of the standards, localization revisions to these standards with current quality improvement programs, and develop a unifying documenting system.

"... *The autonomy to choose to embark or not on the accreditation process and was allowed to manage its accreditation process with independence..., this high autonomy led to the possibility of adjusting the accreditation standards to the hospital's context, which resulted in a high acceptance of the accreditation procedures and positive outcomes in terms of quality and patient safety improvements.*" [32] (p.1253)

With respect to communal appraisal, most professional groups emphasised that the accreditation process was broadly associated with positive changes in hospital performance and quality outcomes. They saw accreditation as a useful tool for improving the routine care they provided to their patients and for overall organisational performance. They also acknowledged

its potential for organising the work processes, enhancing the hospital safety culture, and increasing patients' satisfaction. However, while the accreditation's perceived value was identified as a significant enabler for its implementation and was expected to increase the likelihood of being employed by many health organisations in the future, most groups of professionals were clear that evidence of its effectiveness, while important, was inadequate in itself to support and sustain normalisation into daily practice. Indeed, effective normalisation of accreditation would also necessitate contextual integration, sufficient time, and adequate resources.

"*There is evidence that hospitals rapidly increase compliance with the accreditation standards and improve their organisational processes in the months prior to the surveys, but there is still much less evidence that this brings benefits to the clinical process and the outcome of the healthcare systems.*" [30] (p.12).

A significant level of individual appraisal was also evident, with health professionals highlighting the fact that the accreditation had a positive impact on the quality of care and patient safety, as well as more effective teamwork, which enhances the health system in general.

"*Interviewees felt that hospital accreditation contributed to the improvement of healthcare quality in general, and more specifically to patient safety, as it fostered staff reflection, a higher standardization of practices, and a greater focus on quality improvement.*" [32] (p.1242).

Measuring the benefits or problems of the accreditation, either through formalised and/or informal methods such as key performance indicators (KPIs) or quality measures, enables a formal evaluation of the impact of the accreditation on health organisations.

"*The quality improvement index. . ., showed a significantly greater improvement in the performance of accredited hospitals compared with the control hospitals. These significant improvements. . . were associated with direct cost savings that would benefit both hospitals and the overall health-care system.*" [41] (p.98).

## Discussion, limitations and conclusion

To the best of our knowledge, this is the first comprehensive systematic review that uses the NPT theoretical framework to review existing international literature on the accreditation implementation process and the quality of healthcare services in hospital settings. In this review, data have been interpreted and presented using the four constructs of the NPT to better understand the impact of the accreditation process on the quality of healthcare and to assess the barriers and facilitators to the implementation process in hospital settings. According to the NPT, it is clear that there is a lack of studies concerning critical areas of accreditation implementation work, namely sense-making, involvement, and appraisal work.

***For the coherence construct,*** the findings from the reviewed articles suggest that hospital accreditation is widely used by many healthcare organisations worldwide. Many healthcare professional groups believe it makes sense because of its link to improved hospital performance and healthcare outcomes [36, 42, 43]. Although the accreditation process has many similarities with other quality improvement methods, healthcare professionals have a good understanding of how it differs from other methods of work. However, the specific objectives, possible benefits, and value of accreditation as a tool for quality assurance are not necessarily clear to all groups of health professionals, especially those who are not directly taking part in

the implementation process. There is also a lack of information in the literature about what techniques are best for creating a clear understanding and sharing a vision about the specific objectives, value, and potential benefits of accreditation. Is this something that can be accomplished by providing a variety of training and workshops or by educating health professionals about their roles and responsibilities in the execution of accreditation, or is it more related to role identity and culture, which can be achieved through modification and monitoring? Therefore, the literature still needs to explore that area, specifically with regards to making a collective perception of the specific objectives, potential benefits, and importance of accreditation, which is still challenging across diverse healthcare professional groups, particularly physicians.

Furthermore, there is a clear paucity of research into how healthcare professionals are educated about their specific role in accreditation implementation. While healthcare professionals generally perceived the main goals of accreditation and how they could standardise work processes and enhance the quality of routine patient care, they were concerned with their limited knowledge and inadequate awareness regarding their specific roles and responsibilities in the implementation of accreditation processes. These difficulties were considered barriers to effectively implementing the accreditation process and, in many cases, created negative feelings towards accreditation, which therefore promoted resistance to change among various healthcare professionals. These findings emphasise the significance of further examining health professionals' views on their understanding of their roles and duties in the accreditation process, as well as the effective techniques for doing so.

***In terms of cognitive participation,*** the findings suggest that there is evidence of both implementation (buy-in) and resistance to hospital accreditation among healthcare professionals. The implementation of accreditation is not a straightforward intervention for improving the quality of provided healthcare and hospital performance. Indeed, there are several significant challenges associated with accreditation implementation, particularly with the enrolment and implementation of certain healthcare professional groups, such as senior managers and physicians. Although the findings emphasise the importance of support from accreditation specialists and key persons in the healthcare setting in improving conditions to take forward the accreditation implementation process and make it part of everyday work, there is a clear gap in the literature regarding how healthcare workers managed to get their colleagues engaged in the implementation process and how to increase their enrolment. Thus, it would be interesting to conduct further research to explore the above critical issues and identify strategies for how to encourage and support all healthcare professionals to be actively involved in the accreditation process.

Another challenge linked to accreditation implementation is how various health professional groups think that they are legitimate or have the right to be engaged in the accreditation process and how to change their scepticism about the value and potential benefits of accreditation to improve the quality of health care services [9, 37]. As reported in this review, some groups of health professionals did not take part in the accreditation implementation process as smoothly or effectively as other groups. One reason, according to NPT, could be that the accreditation is described as an interrelated process. This means that a lack of staff awareness about accreditation objectives and implementation requirements (sense-making work) or the fact that not all health professionals received adequate education and training prior to the accreditation process could lead to their scepticism about the value and potential benefits of accreditation and their refusal to participate in the implementation process. However, it is still unclear how health professionals view the exact benefits of accreditation or whether they believe it has legitimate implications. As a result, more research is needed to determine why and how such health professionals can be best supported to achieve an advanced working role in the accreditation process and have a positive perception of its potential outcomes.

*In relation to collective action (enacting work),* almost all of the articles featured content that detailed and described the operational effort that health professionals did to integrate the accreditation standards into their current practises. The findings highlighted the significance of skills for health professionals involved in accreditation execution. Although the accreditation process improved health professionals' abilities, performance, information sharing, and teamwork, it also had a negative impact due to increased workloads and time constraints, which resulted in a degree of role conflict for health professionals. Many groups of health professionals see this role conflict as an impediment to effective participation in the accreditation process. There was also a focus on the importance of adequate resources, such as financial, human, and social resources, and how these resources could encourage health professionals to participate in order to embed accreditation standards in practise and facilitate the successful implementation of accreditation. It does appear that limited resources are problematic as they can inhibit the effective implementation of the accreditation process, and this also raises the issue of the worthiness of participation in accreditation as it is considered a costly investment.

There are a few articles that concern the fourth construct, *reflexive monitoring (appraisal work)*. These revealed that there is a lack of evaluation or auditing of the accreditation process. However, some of the included articles clearly reported and discussed the appraisal work that health professionals did to assess and understand the ways in which the accreditation affects them and other stakeholders and to evaluate the impact and benefits of accreditation on the quality of care and patient safety. Overall, and in line with previous literature [12, 32, 44], health professionals emphasise that the accreditation process is broadly associated with positive change in hospital performance and quality outcomes, as well as contributing to the enhancement of the health system in general. The evaluation process is important since it can help in the enhancement of the accreditation implementation process by identifying, analysing, and, in certain cases, resolving difficulties.

Finally, while some papers in this review found that healthcare professionals with experience of successful implementation of the accreditation process in hospital settings report a range of positive changes (benefits) in providing healthcare for patients that they highly value, other articles revealed inconsistent findings and/or contradicting views among different health professional groups about the impact of accreditation on healthcare services. Therefore, additional research with empirical evidence is needed to strengthen the evidence base regarding the evaluation of the accreditation implementation process and its perceived benefits among healthcare professionals. It would also be beneficial to develop an in-depth understanding of the processes through which accreditation may influence healthcare quality and outcomes.

## Limitations

The study has at least three limitations. While a systematic search of the literature was conducted in accordance with PRISMA guidelines, it is likely that relevant articles were dropped or published after the initial search. However, the reviewers reduced this by double screening, searching the reference lists of the articles, and conducting an update search before the study's conclusion. Second, there is the possibility of researcher bias when interpreting qualitative data. Thus, a concerted effort was made to achieve the highest possible degree of objectivity through a combination of reflection, meticulous implementation of a transparent interpretation process, and assessing the veracity of the findings in light of the international work. Third, using a theoretical framework to structure the synthesis and interpret the study results raises the possibility that authors will be constrained by theory and overlook important data. However, only a few data points in this study could not be connected to the NPT constructs, and those data points were either too general and difficult to understand without context, or they were not directly relevant to the research objectives [45].

## Conclusion

The application of NPT enabled the identification of the key factors that inform and influence the successful implementation of accreditation within the hospital setting. While there are inconsistent findings on the impact of accreditation on improving the quality of healthcare services, with mixed views among healthcare professionals and scant evidence about its effectiveness, accreditation continues to gain acceptance internationally as a quality assurance tool to support best practices in evaluating the quality outcomes of healthcare delivered. This study provides valuable insights into the critical success factors that must be considered in the implementation of accreditation. Policymakers, healthcare organisations, and researchers should examine these factors in advance to ensure the successful implementation and sustainability of accreditation programmes within the hospital setting. As a result, our work contributes to both the NPT literature and hospital accreditation. Future research should also promote the use of a theoretical framework as a guide to improve understanding and help bridge the gap between healthcare research and the implementation of new interventions.

## Supporting information

**S1 File. PRISMA checklist for systematic review.**
(DOCX)

**S2 File. A complete description of the result of electronic search strategies of two databases (Ovid MEDLINE and PUBMED).**
(DOCX)

**S3 File. Data extraction sheet for potentially relevant studies on full-text screening level.**
(DOCX)

**S4 File. Normalization Process Theory (NPT) coding framework used for analysis of review data on hospital accreditation implementation.**
(DOCX)

**S5 File. Coding of data from the included studies into the NPT framework using a Microsoft Excel Worksheet.**
(XLSX)

**S6 File. The quality assessment tool for included articles drawn from Hawker et al., 2002.**
(DOCX)

**S7 File. The included and excluded studies and reasons for exclusion.**
(DOCX)

**S8 File. Summary of all articles included in the final review (n = 21).**
(DOCX)

**S9 File. Additional references: Included studies in the systematic review but did not cited in the text.**
(DOCX)

## Author Contributions

**Conceptualization:** Mohammad J. Alhawajreh, Audrey S. Paterson, William J. Jackson.

**Data curation:** Mohammad J. Alhawajreh, Audrey S. Paterson, William J. Jackson.

**Formal analysis:** Mohammad J. Alhawajreh, Audrey S. Paterson, William J. Jackson.

**Methodology:** Mohammad J. Alhawajreh, Audrey S. Paterson, William J. Jackson.

**Project administration:** Mohammad J. Alhawajreh.

**Resources:** Mohammad J. Alhawajreh, Audrey S. Paterson, William J. Jackson.

**Software:** Mohammad J. Alhawajreh, Audrey S. Paterson, William J. Jackson.

**Supervision:** Audrey S. Paterson, William J. Jackson.

**Validation:** Mohammad J. Alhawajreh, Audrey S. Paterson, William J. Jackson.

**Writing – original draft:** Mohammad J. Alhawajreh, Audrey S. Paterson, William J. Jackson.

**Writing – review & editing:** Mohammad J. Alhawajreh, Audrey S. Paterson, William J. Jackson.

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
