## [Decision Letter · Decision Letter 0]

31 Aug 2023

PONE-D-23-09325Impact of hospital accreditation on quality improvement in healthcare: A systematic reviewPLOS ONE

Dear Dr. Alhawajreh,

Thank you for submitting your manuscript to PLOS ONE. After careful consideration, we feel that it has merit but does not fully meet PLOS ONE’s publication criteria as it currently stands. Therefore, we invite you to submit a revised version of the manuscript that addresses the points raised during the review process.

We look forward to receiving your revised manuscript.

Kind regards,

Tai Ming Wut

Academic Editor

PLOS ONE

Journal Requirements:

"NO authors have competing interests"

Reviewers' comments:

Reviewer's Responses to Questions

**Comments to the Author**

1. Is the manuscript technically sound, and do the data support the conclusions?

Reviewer #1: Partly

2. Has the statistical analysis been performed appropriately and rigorously? 

Reviewer #1: I Don't Know

3. Have the authors made all data underlying the findings in their manuscript fully available?

Reviewer #1: No

4. Is the manuscript presented in an intelligible fashion and written in standard English?

Reviewer #1: Yes

5. Review Comments to the Author

Reviewer #1: This is an important paper and review. It is important that there is clarity in the reviewing processes.

1. What is the objective for the review as two versions exist?

Objective. To build the evidence for the impact of accreditation on quality improvement of healthcare services and develop an understanding of the contextual factors influencing

accreditation implementation in the hospital setting through the lens of Normalisation Process Theory (NPT)

This study provides a comprehensive systematic review to identify contextual factors that may influence hospital accreditation implementation and develop a better understanding of their potential implications

2. There is no PICO set out.

3) There is no definition of quality improvement or how impact is defined or measured.

4) What setting of healthcare are included or excluded.

5) The reviewing processes - screening and abstraction - how many authors were involved as Cochrane states 2 independent reviewers and extraction in duplicate

6) Why was a qualitative synthesis used when 9 studies used qualitative methods and 4 used mixed methods. How were quantitative data reported?

7) SWiM should be used and referenced https://www.bmj.com/content/368/bmj.l6890

8) Data 'were' - sometimes data 'was' is written

The definitions used are critical to presenting this review and what studies were included or excluded.

6. PLOS authors have the option to publish the peer review history of their article (what does this mean?). If published, this will include your full peer review and any attached files.

Reviewer #1: No

---

## [Author Response · Author response to Decision Letter 0]

10 Oct 2023

Response to Reviewers

Title: Impact of hospital accreditation on quality improvement in healthcare: A systematic review

Version: 2

Date: Oct 08, 2023 

Author's response to reviews: see over

Subject: Response to reviewer comments for the manuscript “Impact of hospital accreditation on quality improvement in healthcare: A systematic review.” 

Thank you for the opportunity to revise and resubmit the above manuscript to your esteemed journal. We thank the academic editor and reviewer(s) for their detailed and thorough comments and have addressed their concerns in the revised version of the manuscript. Kindly find below a point-by-point description of the changes made to the manuscript.

Academic Editor

Comment 1:

Response 1:

Thank you for highlighting these changes, this has now been completed.

Comment 2:

"NO authors have competing interests"

Response 2:

I appreciate this, I have included this in the cover letter.

Reviewer comment:

Reviewer's Responses to Questions

Comments to the Author 

1. Is the manuscript technically sound, and do the data support the conclusions?

Reviewer #1: Partly

Response:

Many sections within the manuscript have been modified, and more detailed information has been added in order to make it clearer for readers. These changes can be tracked easily in the "Revised Manuscript with Track Changes File.".

2. Has the statistical analysis been performed appropriately and rigorously?

Reviewer #1: I Don't Know

Response:

The study is a qualitative systematic review, so it has no statistical analysis.

3. Have the authors made all data underlying the findings in their manuscript fully available?

Reviewer #1: No

Response:

Thank you for your comment. 

• A summary of the 21 articles included in the final review and the quality assessment results were previously presented in supporting information files (S6 File). 

• Per your suggestion, we have uploaded a separate file of included and excluded studies and reasons for exclusion.

• For each of the 21 articles included in the qualitative analysis; review data on the facilitators and barriers to, and the impact of hospital accreditation on quality improvement of healthcare services was extracted, summarized, and coded to the related construct of the NPT framework using a Microsoft Excel Worksheet (the file uploaded separately).

4. Is the manuscript presented in an intelligible fashion and written in standard English?

Reviewer #1: Yes

Response:

Thank you.

5. Review Comments to the Author

Reviewer #1: 

This is an important paper and review. It is important that there is clarity in the reviewing processes.

Response:

Thank you for your kind review. We really appreciate the time and effort which has been taken to review this manuscript.

Comment 1:

1. What is the objective for the review as two versions exist?

Objective. To build the evidence for the impact of accreditation on quality improvement of healthcare services and develop an understanding of the contextual factors influencing accreditation implementation in the hospital setting through the lens of Normalisation Process Theory (NPT)

This study provides a comprehensive systematic review to identify contextual factors that may influence hospital accreditation implementation and develop a better understanding of their potential implications

Response 1:

Thank you for your comment, the objective was modified in a clearer way throughout the paper; lines 31-34 and lines 113-121.

Comment 2:

2. There is no PICOS set out.

Response 2:

The PICOS criteria were previously included in table 3 A Complete Descriptions of Inclusion/ Exclusion Criteria for Articles on Accreditation Implementation, however we have made this clearer; we have added a sentence “see table 3 below”; lines 199-201 and written the PICOS criteria in bold within table 3; page 10.

Comment 3:

3) There is no definition of quality improvement or how impact is defined or measured.

Response 3:

Per your suggestion we have added a definition of healthcare quality including explanation of its measures; lines 77-87.

Comment 4:

4) What setting of healthcare are included or excluded.

Response 4:

The included / excluded healthcare settings were previously mentioned in table 3 A Complete Descriptions of Inclusion/ Exclusion Criteria for Articles on Accreditation Implementation; 

• Under Inclusion Criteria point No. 2. Article conducted in hospital settings (the Population).

• Under Exclusion Criteria point No. 3. Article where the population studied was non-hospital setting (e.g., primary healthcare organizations or other community-based healthcare organizations).

, however, we have made this clearer by writing it in bold following your comments; page 10.

Comment 5:

5) The reviewing processes - screening and abstraction - how many authors were involved as Cochrane states 2 independent reviewers and extraction in duplicate

Response 5:

Thank you for this comment. The section on selection criteria and the data collection process was rewritten to be more explicit, explaining the number of reviewers for both screening eligibility and data extraction from included articles; lines 203-204, lines 209-211, lines 215-216, lines 244-248, and lines 260-261.

Comment 6:

6) Why was a qualitative synthesis used when 9 studies used qualitative methods and 4 used mixed methods. How were quantitative data reported?

Response 6:

Explaining the reasons behind using qualitative synthesis and the difficulty of conducting a meta-analysis had previously been partially included within the Synthesis of Results section; however, we have made this clearer following your comments; lines 226-235.

For the quantitative studies, the coding and interpretation process of review data focused mainly on the results and discussion parts of each included article. The findings were first coded and brought together in a spreadsheet to better manage the data. Open coding was conducted first, where findings were broken into chunks that relate to different concepts or ideas. Axial coding was then conducted, which involves organizing the emerging concepts into themes. Themes were pre-identified based on the study objectives; lines 244-248 and lines 228-235.

Comment 7:

7) SWiM should be used and referenced https://www.bmj.com/content/368/bmj.l6890

Response 7:

Thank you for your comment. Our systematic review followed the current best practice approach outlined in the Preferred Reporting Items for Systematic Reviews and Meta-Analyses (PRISMA) statement, and the Cochrane Handbook for Systematic Reviews of Interventions (referenced).

In addition, a concerted effort was made to achieve the highest possible degree of objectivity through a combination of reflection; detailed description of the methods used; meticulous implementation of a transparent interpretation process, assessing the veracity of the findings in light of the international work; clear links between the included data, the synthesis, and the conclusions through the theoretical framework of the Normalisation Process Theory, and sufficient reporting of the limitations of the synthesis.

So, we do think we conducted the narrative synthesis with a high degree of clarity and transparency, and we reported this systematic review considering the best practices mentioned above.

Comment 8:

8) Data 'were' - sometimes data 'was' is written

Response 8:

Thank you for your comment. The use of “was or were” with “data” was reviewed throughout the study and modified accordingly.

Comment:

The definitions used are critical to presenting this review and what studies were included or excluded.

Response:

Per your suggestion, we have uploaded a separate file of included and excluded studies and reasons for exclusion.

Thank you for providing us with the opportunity to address the reviewers’ comments. We look forward to the outcome of the review process. 

Thank you,

Authors

---

## [Decision Letter · Decision Letter 1]

27 Oct 2023

Impact of hospital accreditation on quality improvement in healthcare: A systematic review

PONE-D-23-09325R1

Dear Mohammad J. Alhawajreh,

We’re pleased to inform you that your manuscript has been judged scientifically suitable for publication and will be formally accepted for publication once it meets all outstanding technical requirements.

Kind regards,

Tai Ming Wut

Academic Editor

PLOS ONE

Additional Editor Comments (optional):

Reviewers' comments:

Reviewer's Responses to Questions

**Comments to the Author**

1. If the authors have adequately addressed your comments raised in a previous round of review and you feel that this manuscript is now acceptable for publication, you may indicate that here to bypass the “Comments to the Author” section, enter your conflict of interest statement in the “Confidential to Editor” section, and submit your "Accept" recommendation.

Reviewer #1: All comments have been addressed

2. Is the manuscript technically sound, and do the data support the conclusions?

Reviewer #1: Yes

3. Has the statistical analysis been performed appropriately and rigorously? 

Reviewer #1: N/A

4. Have the authors made all data underlying the findings in their manuscript fully available?

Reviewer #1: Yes

5. Is the manuscript presented in an intelligible fashion and written in standard English?

Reviewer #1: Yes

6. Review Comments to the Author

Reviewer #1: (No Response)

7. PLOS authors have the option to publish the peer review history of their article (what does this mean?). If published, this will include your full peer review and any attached files.

Reviewer #1: No

---

## [Editor Report · Acceptance letter]

23 Nov 2023

PONE-D-23-09325R1 

*Impact of hospital accreditation on quality improvement in healthcare: A systematic review*

Dear Dr. Alhawajreh:

I'm pleased to inform you that your manuscript has been deemed suitable for publication in PLOS ONE. Congratulations! Your manuscript is now with our production department. 

Kind regards, 

on behalf of

Dr. Tai Ming Wut 

Academic Editor

PLOS ONE